# PARAMETRIZING FILTERS OF A CNN WITH A GAN

## ABSTRACT

It is commonly agreed that the use of relevant invariances as a good statistical bias is important in machine-learning. However, most approaches that explicitly incorporate invariances into a model architecture only make use of very simple transformations, such as translations and rotations. Hence, there is a need for methods to model and extract richer transformations that capture much higher-level invariances. To that end, we introduce a tool allowing to parametrize the set of filters of a trained convolutional neural network with the latent space of a generative adversarial network. We then show that the method can capture highly non-linear invariances of the data by visualizing their effect in the data space.

## 1 INTRODUCTION

In machine-learning, solving a classification task typically consists of finding a function $f : \mathcal{X} \to \mathcal{Y}$, from a rather large data space $\mathcal{X}$ to a much smaller space of labels $\mathcal{Y}$. Such a function will therefore necessarily be invariant to a lot of transformations of its input data. It is now clear that being able to characterize such transformations can greatly help the learning procedure, one of the most striking examples being perhaps the use of convolutional neural networks (CNN) for image classification (Krizhevsky et al., 2012), with built-in translation invariance via convolutions and subsequent pooling operations. But as a convolutional layer is essentially a fully connected layer with a constraint tying some of its weights together (LeCun et al., 1995), one could expect other invariances to be encoded in its weights after training. Indeed, from an empirical perspective, CNNs have been observed to naturally learn more invariant features with depth (Goodfellow et al., 2009; Lenc & Vedaldi, 2015), and from a theoretical perspective, it has been proven that under some conditions satisfied by the weights of a convolutional layer, this layer could be re-indexed as performing a convolution over a bigger group of transformations than only translations (Mallat, 2016).

It is exciting to note that there has recently been a lot of interest in theoretically extending such successful invariant computational structures to general groups of transformations, notably with group invariant scattering operators (Mallat, 2012), deep symmetry networks (Gens & Domingos, 2014), group invariant signal signatures (Anselmi et al., 2015), group invariant kernels (Mroueh et al., 2015) and group equivariant convolutional networks (Cohen & Welling, 2016). However, practical applications of these have mostly remained limited to linear and affine transformations. Indeed, it is a challenge in itself to parametrize more complicated, non-linear transformations preserving labels, especially as they need to depend on the dataset. In this work, we seek to answer this fundamental question:

*What invariances in the data has a CNN learned during its training on a classification task and how can we extract and parameterize them?*

The following is a brief summary of our method: Considering an already trained CNN on a labeled dataset, we train a generative adversarial network (GAN) (Goodfellow et al., 2014) to produce *filters* of a given layer of this CNN, such that the filters' convolution output be indistinguishable from the one obtained with the real CNN. We combine this with an InfoGAN (Chen et al., 2016) discriminator to prevent the generator from producing always the same filters. As a result, the generator provides us with a smooth, data-dependent, non-trivial parametrization of the set of filters of this CNN, characterizing complicated transformations irrelevant for this classification task. Finally, we describe how to visualize what these smooth transformations of the filters would correspond to in the image space.

## 2 BACKGROUND

### 2.1 GENERATIVE ADVERSATIAL NETWORKS

A Generative Adversarial Network (Goodfellow et al., 2014) consists of two players, the generator $G$ and the discriminator $D$, playing a minimax game in which $G$ tries to produce samples indistinguishable from some given true distribution $p_{data}$, and $D$ tries to distinguish between real and generated samples. $G$ typically maps a random noise $z$ to a generated sample $G(z)$, transforming the noise distribution into a distribution $p_g$ supposed to match $p_{data}$. The objective function of this minimax game with maximum likelihood is given by

$$\min_G \max_D V(D, G) = \mathbb{E}_{x \sim p_{data}}[\log D(x)] + \mathbb{E}_z[\log(1 - D(G(z)))].$$

The noise space, input of the generator, is also called its latent space.

### 2.2 INFORMATION MAXIMIZING GENERATIVE ADVERSARIAL NETS

In an InfoGAN (Chen et al., 2016), the generator takes as input not only the noise $z$ but also another variable $c$, called the latent code. The aim is to make the generated samples $G(z, c)$ depend on $c := (c_1, ..., c_n)$ in a structured way, for instance by choosing independent $c_i$'s, modelling $p(c)$ as $\prod_i p(c_i)$. In order to avoid a trivial correspondence between $c$ and $G(z, c)$, the InfoGAN procedure maximizes the mutual information $I(c, G(z, c))$.

The mutual information $I(X, Y)$ between two random variables, defined with an entropy $H$ as $I(X, Y) := H(X) - H(X|Y) = H(Y) - H(Y|X)$ is symmetric, measures the amount of information that is known about the value of one random variable when the value of the other one is known, and is equal to zero when they are independent. Hence, maximizing the mutual information $I(c, G(z, c))$ prevents $G(z, c)$ from being independent of $c$.

In practice, $I(c, G(z, c))$ is indirectly maximized using a variational lower bound

$$L_I(G, Q) \leq I(c, G(z, c)),$$

where $Q(c|x)$ approximates $P(c|x)$ and

$$L_I(G, Q) := \mathbb{E}_{c \sim p(c), x \sim G(z,c)}[\log Q(c|x)] + H(c).$$

The minimax game becomes

$$\min_{G,Q} \max_D V_I(D, G, Q) = V(D, G) - \lambda L_I(G, Q),$$

where $\lambda$ is a hyperparameter controlling the mutual information regularization.

## 3 EXTRACTING INVARIANCES BY LEARNING FILTERS

Let CNN be an already trained CNN, whose $\ell^{th}$-layer representation of an image $I$ will be denoted by $\text{CNN}_\ell(I)$, with $\text{CNN}_0(I) = I$. As our goal is to learn what kind of filters such a CNN would learn at layer $\ell$, it could be tempting to simply train a GAN to match the distribution of filters of this CNN's layer. However, this set of filters is way too small of a dataset to train a GAN $(G, D)$, which would cause the generator $G$ to massively overfit instead of extracting the smooth, hidden structure lying behind our discrete set of filters. To cope with this problem, instead of feeding the discriminator $D$ alternatively with filters produced by $G$ and real filters from $\text{CNN}_\ell$, we propose to feed $D$ with the *activations* of these filters caused by the data passing through the CNN, *i.e.* alternatively with $\text{CNN}_\ell(I)$ or $Conv(\text{CNN}_{\ell-1}(I), G(z))$, corresponding respectively to real and fake samples. Here, $I$ is an image sampled from data, $z$ is sampled from the latent space of $G$ and $Conv(\text{CNN}_{\ell-1}(I), G(z))$ is the activation obtained by passing $I$ through each layer of CNN but while replacing the filters of the $\ell^{th}$-layer by $G(z)$.

In short, in each step, the generator $G$ produces a set of filters for the $\ell^{th}$-layer of the CNN. Next, different samples of data are passed through one CNN using its *real* filters and through the same CNN, but having its $\ell^{th}$-layer filters replaced by the *fake* filters produced by $G$. The discriminator

$D$ will then try to guess if the activation it is fed was produced using real or generated filters at the $\ell^{th}$-layer, while the generator $G$ will try to produce filters making the subsequent activations indistinguishable from some obtained with real filters.

However, even though this formulation allows us to train the GAN on a dataset of reasonable size, saving us from an otherwise unavoidable overfitting, $G$ could a priori still always produce the same set of filters to fool $D$. Ideally it simply reproduces the real filters of $\texttt{CNN}_\ell$. To overcome this problem, we augment our model by an InfoGAN discriminator whose goal will be to predict which noise $z$ was used to produce $Conv(\texttt{CNN}_{\ell-1}(I), G(z))$. This prevents $G$ from always producing the same filters, by preventing $z$ and $G(z)$ from being indenpendent random variables. Note that, just as the GAN discriminator, the InfoGAN discriminator does not act directly on the output of $G$ - the filters - but on the activation output that these filters produce.

In this setting, the noise $z$ of the generator $G$ plays the role of the latent code $c$ of the InfoGAN. As in the original InfoGAN paper (Chen et al., 2016), we train a neural network $Q$ to predict the latent code $z$, which shares all its layers but the last one with the discriminator $D$. Finally, by modelling the latent codes as independent gaussian random variables, the term $L_I(G, Q)$ in the variational bound being a log-likelihood, it is actually given by an $L^2$-reconstruction error. The joint training of these three neural networks is described in Algorithm 1 and illustrated in Figure 1.

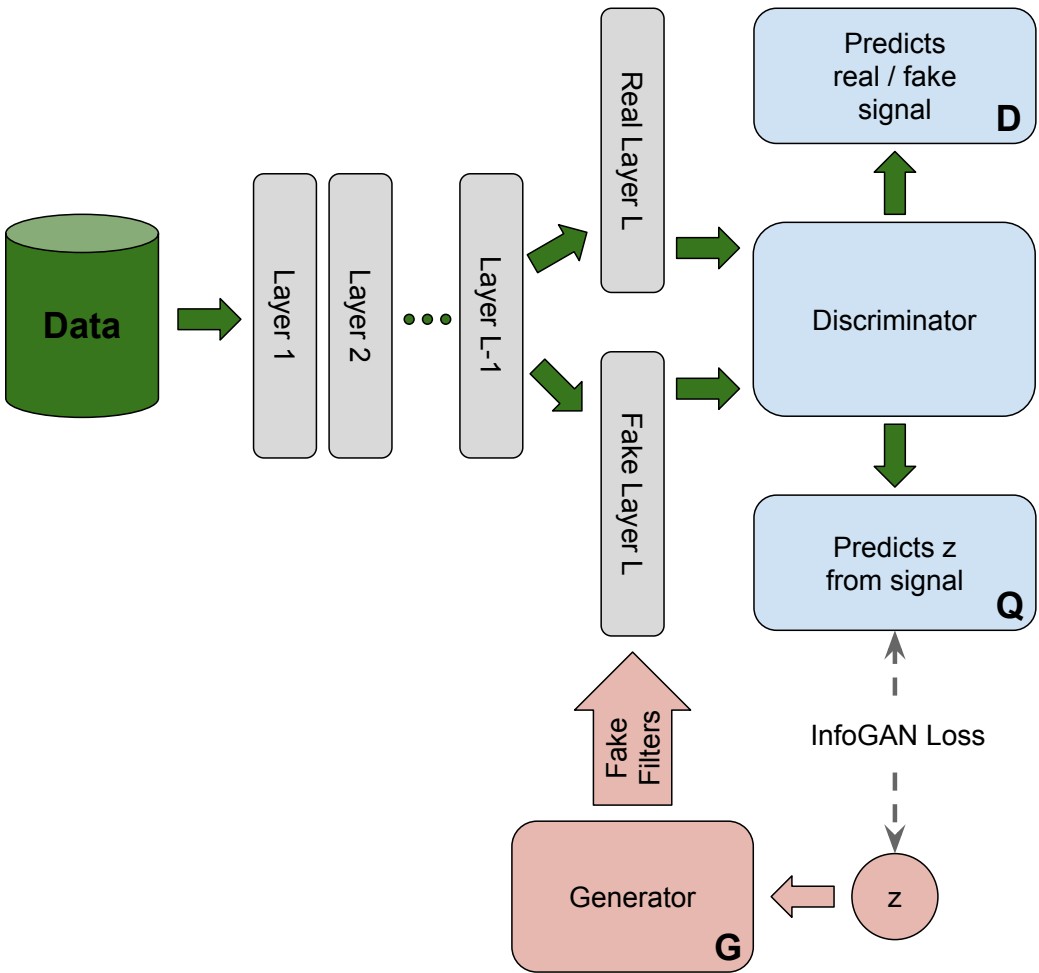

Figure 1: Illustration of how the different neural networks interact with each other. CNN layers are depicted in light gray. The flow of data is shown in green, while the generation of the filters by the generative model is shown in red. The discriminator part of the GAN is shown in blue. Note that the discriminator does not have direct access to the generated filters, but can only observe the data after it has passed through them. The CNN is fixed, while $G$, $D$ and $Q$ are trained jointly.

---

**Algorithm 1** Minibatch stochastic gradient descent training of $D$, $G$ and $Q$.

1: **for** number of training iterations **do**
2:     • Sample minibatch of $m$ noise samples $\{z^{(1)}, ..., z^{(m)}\}$ from noise prior $p_{noise}(z)$.
3:     • Generate the filters $G_{\theta_g}(z^{(1)}), ..., G_{\theta_g}(z^{(m)})$.
4:     • Sample minibatch of $m$ examples $\{I^{(1)}, ..., I^{(m)}\}$ from data distribution $p_{data}(I)$.
5:     • Pass the data through the CNN with the real and generated filters, to obtain the $\texttt{CNN}_\ell(I^{(i)})$'s and $Conv(\texttt{CNN}_{\ell-1}(I^{(i)}), G_{\theta_g}(z^{(i)}))$'s respectively.
6:     • Feed these to $D_{\theta_d}$ and $Q_{\theta_q}$, letting $D_{\theta_d}$ guess if it was fed $\texttt{CNN}_\ell(I^{(i)})$ or $Conv(\texttt{CNN}_{\ell-1}(I^{(i)}), G_{\theta_g}(z^{(i)}))$, and letting $Q_{\theta_q}$ recover the $z^{(i)}$.
7:     • Update the discriminator by ascending its stochastic gradient:

$$\nabla_{\theta_d} \frac{1}{m} \sum_{i=1}^m [\log D_{\theta_d}(\texttt{CNN}_\ell(I^{(i)})) + \log(1 - D_{\theta_d}(Conv(\texttt{CNN}_{\ell-1}(I^{(i)}), G_{\theta_g}(z^{(i)}))))].$$

8:     • Update the generator by descending its stochastic gradient:

$$\nabla_{\theta_g} \frac{1}{m} \sum_{i=1}^m [\log(1 - D_{\theta_d}(Conv(\texttt{CNN}_{\ell-1}(I^{(i)}), G_{\theta_g}(z^{(i)}))))$$
$$+ \lambda \|z^{(i)} - Q_{\theta_q}(Conv(\texttt{CNN}_{\ell-1}(I^{(i)}), G_{\theta_g}(z^{(i)})))\|_2^2].$$

9:     • Update the InfoGAN discriminator by descending its stochastic gradient:

$$\nabla_{\theta_q} \left( \frac{\lambda}{m} \sum_{i=1}^m \|z^{(i)} - Q_{\theta_q}(Conv(\texttt{CNN}_{\ell-1}(I^{(i)}), G_{\theta_g}(z^{(i)})))\|_2^2 \right).$$

    **end for**
10: The gradient-based updates can use any standard gradient-based learning rule. We used RM-Sprop in our experiments.

---

## 4 VISUALIZING THE LEARNED TRANSFORMATIONS

Using our method, we can parameterize the filters of a trained CNN and thus characterize its learned invariances. But in order to assess what has actually been learned, we need a way to visualize these invariances once the GAN has been trained. More specifically, given a data sample $x$, we would like to know what transformations of $x$ the CNN regards as being invariant. We do this in the following manner:

We take some latent noise vector $z$ and obtain its generated filters $G(z)$. Using those filters, we pass the data sample $x$ through the network to obtain $Conv(\texttt{CNN}_{\ell-1}(x), G_{\theta_g}(z)) =: a_{(x|z)}$, which we call the *activation profile* of $x$ given $z$.

Next we select two dimensions $i$ and $j$ of $z$ at random and construct a *grid* of noise vectors $\{z_k | z_k^i \in [-\xi, \xi], z_k^j \in [-\xi, \xi]\}_k$ by moving around $z$ in the dimensions $i$ and $j$ in a small neighborhood.

For each $z_k$, we use Gradient Descent to start from $x$ and find the data point $x'_k$ that gives the *same activation profile* for the filters generated using $z_k$ as the data point $x$ gave for the filters generated using $z$. Formally, for each $z_k$ we want to find $x'_k$, s.t.

$$x'_k = \underset{x'}{\operatorname{argmin}} \|a_{(x'|z_k)} - a_{(x|z)}\|_2^2 + \Psi(x'),$$

where $\Psi(x')$ is a regularizer corresponding to a *natural image prior*. Specifically, we use the loss function proposed in (Mahendran & Vedaldi, 2015).

By using Gradient Descent and starting from the original data point $x$, we make sure that the solution we find is likely in the neighborhood of $x$, *i.e.* can be obtained by applying a small transformation to $x$.

As a result, from our grid of $z$-vectors, we obtain a grid of $x$-points. This grid in data space represents a *manifold traversal* in a small neighborhood on the manifold of learned invariances. If our method is successful, we expect to see sensible continuous transformations of the original data point along the axes of this grid.

## 5 EXPERIMENTAL RESULTS

We apply our method of extracting invariances on a convolutional neural network trained on the MNIST dataset. In particular, we train a standard architecture featuring 5 convolutional layers with ReLU nonlinearities, max-pooling and batch normalization for 10 epochs on the 10-class classification task.

Once converged, we use our GAN approach to learn the filters of the 4th convolutional layer in the CNN. Since this is one of the last layers in the network, we expect the invariances that we extract to be very high-level and highly nonlinear.

### 5.1 VISUALIZING THE LEARNED INVARIANCES

The results can be seen in Figure 3 and a sample of the learned filters themselves can be seen in Figure 2. Our expectations are clearly met as the resulting outputs are in fact an ensemble of highly nonlinear and high-level transformations of the data. Even more visualizations can be found in the Appendix.

We further hypothesize that if we apply the same method to the filters of one of the first layers in the network, the transformations that we learn will be much more low-level and more pixel-local. To test this, we use our method on the same CNN's second convolutional layer. The results can be seen in Figure 4. As expected, the transformations are much more low-level, such as simple brightness changes or changes to the stroke width.

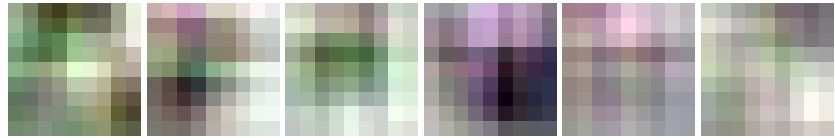

Figure 2: Learned filters of the CNN's 4th layer. We summed one third of the orignal channels together in order to visualize the learned filters.

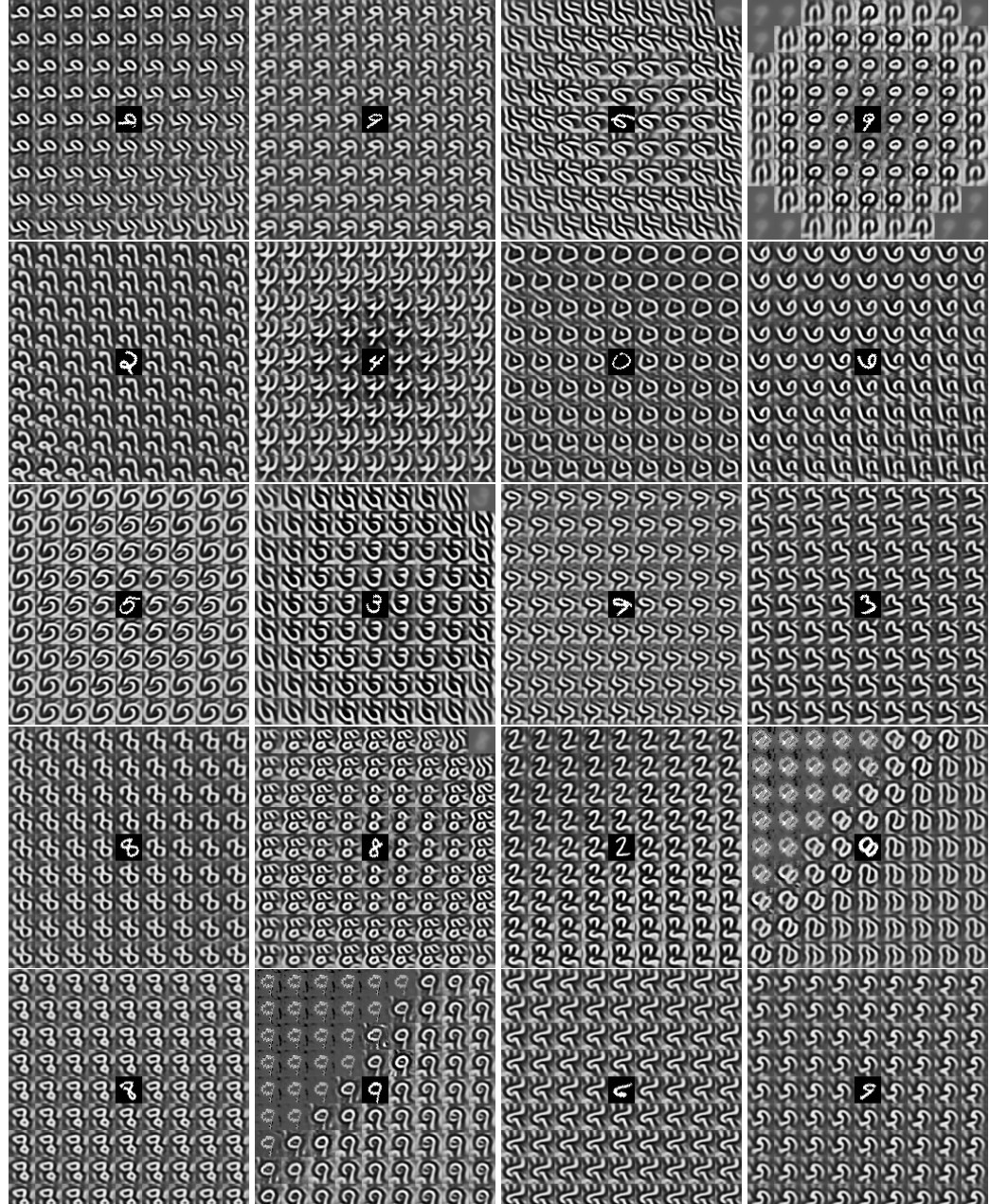

Figure 3: Invariance transformations extracted from the CNN's 4th layer. The middle sample of each grid represents the original data sample, while the rest of the grid are found by matching the original sample's activation profile.

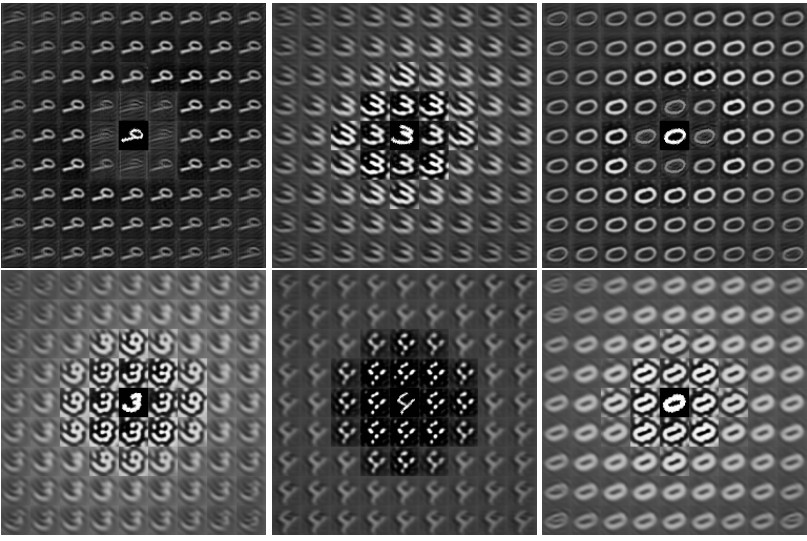

Figure 4: Invariance transformations extracted from the CNN's 2nd layer. The middle sample of each grid represents the original data sample, while the rest of the grid are found by matching the original sample's activation profile.

## 5.2 ASSESSING THE QUALITY OF THE GENERATOR

In order to assess the quality of the generator, we need to be sure that: *(i)* filters produced by the generator would yield a good accuracy on the original classification task of our CNN, and *(ii)* the generator can produce a variety of different filters for different latent noises.

For the first part, we randomly drew 10 noise vectors $z^{(1)}, ..., z^{(10)}$, computed the corresponding set of filters $G(z^{(i)})$ for each of them, and then each data sample $x$, after going through $\text{CNN}_{l-1}$, is passed through each of these 10 $\ell^{th}$-layer and averaged over them, so that the signal fed to the next layer becomes:

$$\frac{1}{10} \sum_{i=1}^{10} Conv(\text{CNN}_{l-1}(x), G(z^{(i)})),$$

all the next layers being re-trained. This averaging can be seen as an average pooling w.r.t. the transformations defined by the generator, which, if the transformations we learned were indeed irrelevant for the classification task, should not induce any loss in accuracy. Our expectations are confirmed, as the test accuracy obtained by following the above procedure is of 0.982, against a test accuracy of 0.971 for the real CNN.

As for the second part, Figure 5 shows a Multi-Dimensional Scaling (MDS) of both the original set of filters of $\text{CNN}_\ell$, and of generated filters for randomly sampled noise vectors. We observe that different noise vectors produce a variety of different filters, which confirms that the generator has not overfitted on the set of real filters. Further, since the generator has learned to produce a variety of filters for each real filter, all the while retaining its classification accuracy, this means that we have truly captured the invariances of the data with regard to the CNN's classification task.

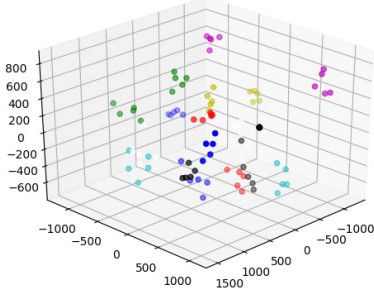

Figure 5: Multi-Dimensional Scaling for the filters produced by the GAN. Individual colors represent different samples for the same filter of the true CNN. The large cluster sizes shows that the GAN is producing a wide variety of different filters for each corresponding real filter.

## 6 CONCLUSION AND FUTURE WORK

Introducing an invariance to irrelevant transformations for a given task is known to constitute a good statistical bias, such as translations in CNNs. Although a lot of work has already been done regarding how to implement known invariances into a computational structure, practical applications of these mostly include very simple linear or affine transformations. Indeed, characterizing more complicated transformations seems to be a challenge in itself.

In this work, we provided a tool allowing to extract transformations w.r.t. which a CNN has been trained to be invariant to, in such a way that these transformations can be both visualized in the image space, and potentially re-used in other computational structures, since they are parametrized by a generator. The generator has been shown to extract a smooth hidden structure lying behind the discrete set of possible filters. It is the first time that a method is proposed to extract the symmetries learned by a CNN in an explicit, parametrized manner.

Applications of this work are likely to include transfer-learning and data augmentation. Future work could apply this method to colored images. As suggested by the last subsection, the parametrization of such irrelevant transformations of the set of filters could also potentially define another type of powerful pooling operation.

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

## A    MORE INVARIANCE VISUALIZATIONS

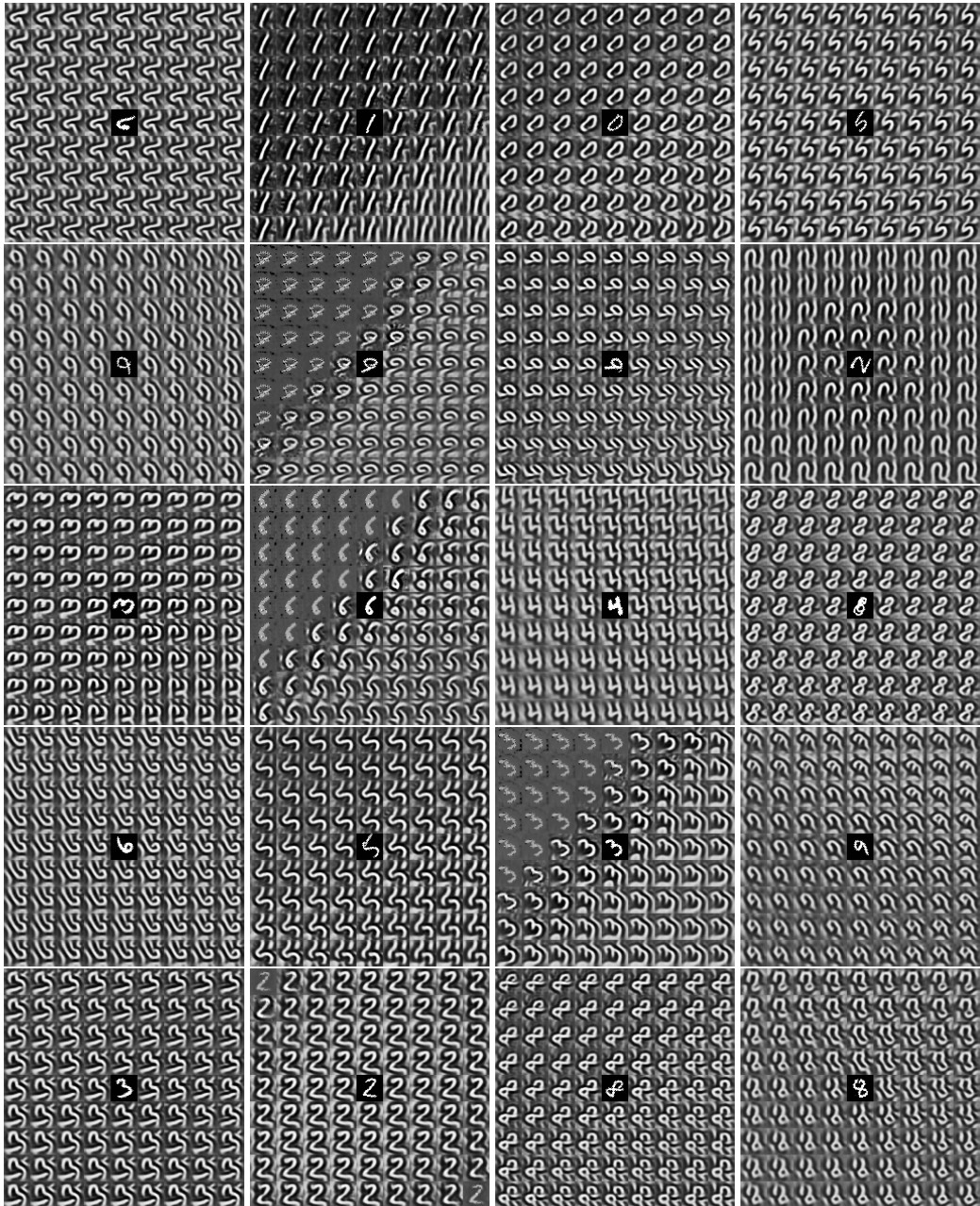

Figure 6: Invariance transformations extracted from the CNN's 4th layer. The middle sample of each grid represents the original data sample, while the rest of the grid are found by matching the original sample's activation profile.

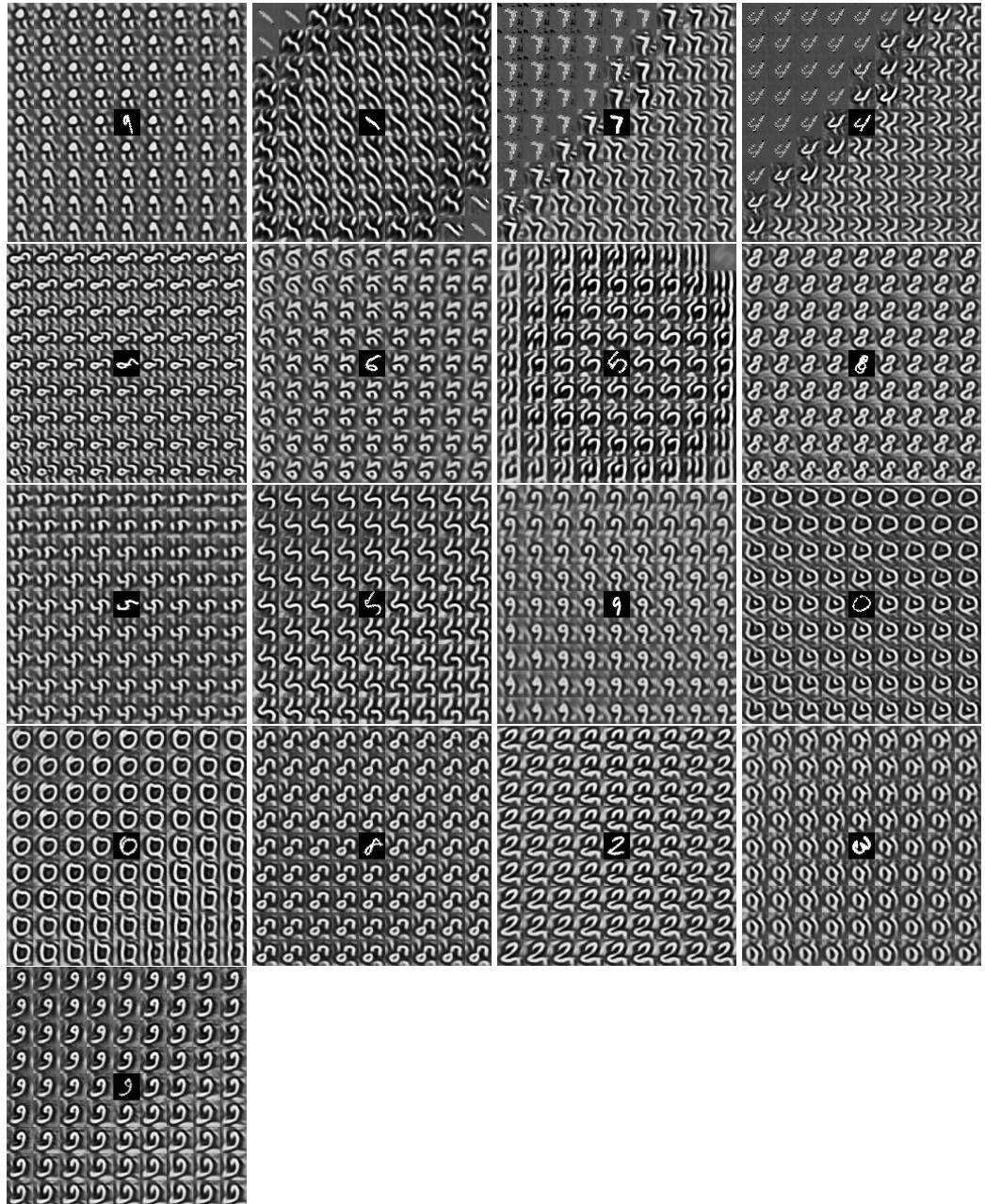

Figure 7: Invariance transformations extracted from the CNN's 4th layer. The middle sample of each grid represents the original data sample, while the rest of the grid are found by matching the original sample's activation profile.

