# OpenReview forum: "Parametrizing filters of a CNN with a GAN"
_ICLR.cc/2018/Conference — Reject_

### Official Review · AnonReviewer1 · 2017-11-25
**Reject - very weak experiments**

**Rating:** 2
**Confidence:** 4

**Review:**

The paper proposes an approach to learning a distribution over filters of a CNN. The method is based on a adversarial training: the generator produces filters, and the discriminator aims to distinguish the activation maps produced by real filters from those produced by the generated ones.

Pros:
1) The general task of learning distributions over network weights is interesting
2) To my knowledge, the proposed approach is new

Cons:
1) Experimental evaluation is very substandard. The experiments on invariances seem to be the highlight of the paper, but they basically do not tell me anything.
 - Figures 3 and 4 take 2 pages, but what should one see there?
 - There are no quantitative results. Could there be a way to measure the invariances?
 - Can the results be applied to some practical task? Why are the results interesting and/or useful?
2) The experiments are restricted to a single dataset - MNIST.  The authors mention that “the test accuracy obtained by following the above procedure is of 0.982, against a test accuracy of 0.971 for the real CNN” - these are very poor accuracies for MNIST. So even the MNIST results do not seem convincing.
3) Presentation is suboptimal, and many details are missing. For instance, architectures of networks are not provided.

To conclude, while the general direction is interesting and the proposed method might work, the experimental evaluation is very poor, and the paper absolutely cannot be accepted for publication.

---

### Official Review · AnonReviewer3 · 2017-11-26
**Lacks control experiments; no new insights**

**Rating:** 4
**Confidence:** 4

**Review:**

Recent work on incorporating prior knowledge about invariances into neural networks suggests that the feature dimension in a stack of feature maps has some kind of group or manifold structure, similar to how the spatial axes form a plane. This paper proposes a method to uncover this structure from the filters of a trained ConvNet. The method uses an InfoGAN to learn the distribution of filters. By varying the latent variables of the GAN, one can traverse the manifold of filters. The effect of moving over the manifold can be visualized by optimizing an input image to produce the same activation profile when using a perturbed synthesized filter as when using an unperturbed synthesized filter.

The idea of empirically studying the manifold / topological / group structure in the space of filters is interesting. A priori, using a GAN to model a relatively small number of filters seems problematic due to overfitting, but the authors show that their InfoGAN approach seems to work well.

My main concerns are:

Controls
To generate the visualizations, two coordinates in the latent space are varied, and for each variation, a figure is produced. To figure out if the GAN is adding anything, it would be nice to see what would happen if you varied individual coordinates in the filter space ("x-space" of the GAN), or varied the magnitude of filters or filter planes. Since the visualizations are as much a function of the previous layers as they are a function of the filters in layer l which are modelled by the GAN, I would expect to see similar plots for these baselines.

Lack of new Insights
The visualizations produced in this paper are interesting to look at, but it is not clear what they tell us, other than "something non-trivial is going on in these networks". In fact, it is not even clear that the transformations being visualized are indeed non-linear in pixel space (note that even a 2D diffeomorphism, which is a non-linear map on R^2, is a linear operator on the space of *functions* on R^2, i.e. on the space of images). In any case, no attempt is made to analyze the results, or provide new insights into the computations performed by a trained ConvNet.

Interpretation
This is a minor point, but I would not say (as the paper does) that the method captures the invariances learned by the model, but rather that it aims to show the variability captured by the model. A ReLU net is only invariant to changes that are mapped to zero by the ReLU, or that end up in the kernel of one of the linear layers. The presented method does not consider this and hence does not analyze invariances.

Minor issues:
- In the last equation on page 2, the right-hand side is missing a "min max".

---

### Official Review · AnonReviewer2 · 2017-11-28

**Rating:** 4
**Confidence:** 5

**Review:**

This paper wants to probe the non-linear invariances learnt by CNNs. This is attempted by selecting a particular layer, and modelling the space of filters that result in activations that are indistinguishable from activations generated by the real filters (using a GAN). For a GAN noise vector a plausible filter set is created, and for a data sample a set of plausible activations are computed. If the noise vector is perturbed and a new plausible filter set is created, the input data can be optimised to find the input that produces the same set of activations. The claim is that the found input represents the non-linear transformations that the layer is invariant to.

This is a really interesting perspective on probing invariances and should be explored more. I am not convinced that this particular method is showing much information or highlighting anything particularly interesting, but could be refined in the future to do so.

It seems that the generated images are not actually plausible images at all and so not many conclusions can be drawn from this method. Instead of performing the optimisation to find x' have you tried visualising the real data sample that gives the closest activations?

I think you may want to consider minimising ||a(x'|z) - a(x|z_k)|| instead to show that moving from x -> x' is the same as is invariant under the transformation z -> z_k  (and thus the corresponding movement in filter space). This (the space between x and x') I think is more interpretable as the invariance corresponding to the space between z and z_k. Have you tried that?

There is no notion of class invariance, so the GAN can find the space of filters that transform layer inputs into other classes, which may not be desirable. Have you tried conditioning the GAN on class?

Overall I think this method is inventive and shows promise for probing invariances. I'm not convinced the current incarnation is showing anything insightful or useful. It also should be shown on more than a single dataset and for a single network, at the moment this is more of a workshop level paper in terms of breadth and depth of results.

---

### Author Response · Authors · 2018-01-05
**Thank you for the feedback**

We thank all reviewers for their valuable and detailed feedback.
We will incorporate it into the work for future publication.

---

### Decision · Program_Chairs · 2018-01-29
**ICLR 2018 Conference Acceptance Decision**

**Decision:**

Reject

**Comment:**

The experiments are not sufficient to support the claim. The authors plan to improve it for future publication.